# Objective and Subjective Outcomes Following Radiofrequency of Inferior Turbinates in Patients with Sleep-Disordered Breathing

**DOI:** 10.3390/diagnostics14161820

**Published:** 2024-08-21

**Authors:** Alfonso Luca Pendolino, Samit Unadkat, Ryan Chin Taw Cheong, Ankit Patel, Joshua Ferreira, Bruno Scarpa, Peter J. Andrews

**Affiliations:** 1Department of ENT, Royal National ENT & Eastman Dental Hospitals, London WC1E 6DG, UK; s.unadkat@nhs.net (S.U.); ryan.cheong@nhs.net (R.C.T.C.); ankit.patel@nhs.net (A.P.); peter.andrews.10@ucl.ac.uk (P.J.A.); 2Ear Institute, University College London (UCL), London WC1X 8EE, UK; 3University College London (UCL) Medical School, London WC1E 6DE, UK; joshua.ferreira.20@ucl.ac.uk; 4Department of Statistical Sciences and Department of Mathematics Tullio Levi-Civita, University of Padova, 35122 Padova, Italy; bruno.scarpa@unipd.it

**Keywords:** sleep-disordered breathing, nasal obstruction, radiofrequency ablation, sino-nasal outcome test, quality of life

## Abstract

Background: Nasal obstruction is a frequent problem amongst patients with sleep-disordered breathing (SDB). Radiofrequency of the inferior turbinates (RFIT) is commonly utilized for inferior turbinate (IT) reduction but its effectiveness in SDB patients remains unproven. We aim to evaluate long-term objective and subjective nasal, olfactory and sleep outcomes following RFIT in SDB patients. Methods: Patients were assessed at baseline (T0) and at 3 months (T1), 6 months (T2) and 12 months (T3) following RFIT. At T0, T1, T2 and T3, the patients underwent objective assessments of their nasal airways and smell function and an evaluation of their quality-of-life, sinonasal, olfactory and sleep symptoms. Sleep studies were carried out at T0 and T2. Results: Seventeen patients (with a median age of 42 years) underwent RFIT. A statistically significant objective and subjective improvement of the patients’ nasal airways was demonstrated at T1. No other statistically significant changes were observed in the patients’ nasal airways, smell, sleep study parameters or patient-reported outcomes at the other follow-ups. A multivariate analysis confirmed a statistically significant influence of age (older), sex (male), a higher BMI, the presence of septal deviation and the presence of allergic rhinitis in some of the studies’ parameters. A statistically significant objective and subjective improvement of the patients’ nasal airways was confirmed in the fitted model when considering the influence of the available variables. Conclusions: Our study confirms that the benefits of RFIT alone in SDB patients are limited and possibly only in the short-term period. Patient-related variables can potentially influence the final outcomes.

## 1. Introduction

Sleep-disordered breathing (SDB) describes a spectrum of various clinical entities ranging from primary snoring to severe obstructive sleep apnoea (OSA) [1]. Both snoring and OSA exhibit a multilevel phenomenon in which the obstruction can occur at each level of the naso-, oro- and hypopharynx and in different proportions [2,3]. The nose represents the first entry point of the air with nasal obstruction significantly impacting the collapsibility of different segments of the pharyngeal lumen [1,4]. Several large-scale population studies have confirmed that nasal blockage contributes to exacerbate OSA and represents an independent risk factor for OSA [5,6,7]. Moreover, OSA patients with nasal obstruction are at higher risk of continuous positive airway pressure (CPAP) intolerance, which constitutes a significant problem as CPAP treatment is the first-line measure for moderate-to-severe OSA. In addition to that, nasal CPAP itself can lead to alterations in the nasal mucosa, like chronic inflammation and fibrosis [8], which can exacerbate CPAP intolerance in patients with an already existing congested nose.

For all these reasons, treatment of nasal obstruction in SDB patients becomes crucial for symptom relief and/or to improve CPAP tolerance, especially in cases in which nasal obstruction is the main subjective barrier to its use. From an anatomical point of view, septal deviation, nasal valve dysfunction and/or inferior turbinate (IT) hypertrophy are the most common findings in SDB patients with reported nasal blockage [9,10,11]. Rhinitis is the main cause of IT hypertrophy, and in this regard, the link between allergic rhinitis (AR), in particular, and sleep impairment is so close that the ARIA (Allergic Rhinitis and its Impact on Asthma) guidelines have categorized the influence of AR on sleep impairment as moderate to severe [12]. Intranasal corticosteroids represent the main treatment of rhinitis, and several studies have confirmed an improvement of sleep study parameters following the use of nasal corticosteroids [13,14,15]. For refractory cases in which patients are not improving with medical treatment, nasal airway surgery can be offered with the aim to improve nasal breathing and, as a result, sleep quality, snoring and daytime fatigue [16].

The reduction in ITs represents an effective surgical option in cases in which IT hypertrophy is the main driver of nasal obstruction [17,18] and can improve sleep quality in cases of concomitant rhinitis and nasal obstruction [16]. So far, several techniques have been described, and available options include turbinoplasty, turbinate out-fracturing, microdebrider-assisted inferior turbinoplasty, electrocautery with monopolar or bipolar instruments, coblation and radiofrequency [19,20,21]. However, no consensus today exists on which surgical technique is most effective in the long term.

Radiofrequency of the IT (RFIT) is a commonly utilized technique for IT reduction and is able to generate a relatively low level of heat in the sub-mucosal layer of the turbinates, thus preserving overlying mucosal integrity and the mucociliary function of the turbinates [22]. Moreover, it has rare complications and can be performed in clinic under local anaesthesia (LA), making it a quick and very attractive option for the surgical management of IT hypertrophy. Although several studies support RFIT effectiveness in managing nasal obstruction secondary to IT hypertrophy [23,24,25,26], few studies have assessed long-term outcomes using disease-specific validated instruments, especially in patients with SDB. Moreover, most studies have focused on subjective outcomes (patient-reported symptoms) of improved breathing and nasal airflow, whereas studies looking at objective measures remain sparce.

In this prospective study, we aimed to evaluate long-term objective and subjective nasal, olfactory and sleep outcomes following RFIT in patients with SDB and IT hypertrophy refractory to medical treatment.

## 2. Materials and Methods

### 2.1. Study Design

A real-life prospective cohort study was conducted to evaluate the efficacy of RFIT in the treatment of IT hypertrophy in patients with SDB. Patients were assessed at baseline (T0), 3 months (T1), 6 months (T2) and 12 months (T3) following RFIT. Patients were asked not to start any nasal steroids during the follow-up period. At the end of T3, patients were discharged to their general practitioners or reassessed in cases of persisting symptoms. Our primary outcome was the improvement of nasal airways following RFIT as measured by peak nasal inspiratory flow (PNIF) and acoustic rhinometry (AR). Secondary outcomes instead were the improvement of sense of smell, sleep symptoms/scores and health-related quality of life (HRQoL) following the procedure (see Section 2.4. on methods used to evaluate olfaction and patient-reported outcome measures).

The study was conducted in accordance with the 1996 Helsinki Declaration. This present study is a retrospective evaluation of service for our department, utilizing anonymized data reviewed in full accordance with national information governance protocols and, thus, did not require separate research ethics committee approval.

### 2.2. Participants’ Characteristics

We included patients with SDB who underwent RFIT under LA for IT hypertrophy between June 2021 and January 2022 at the Royal National Ear, Nose and Throat Hospital (University College London Hospitals, London, UK). Data were collected on demographics, type of sleep disorder (snoring or OSA only, or both), type of rhinitis (allergic vs. non allergic), smoking status, comorbidities, routine medications taken and history of upper airway surgery. Findings at nasal endoscopy and results of skin prick test for common aeroallergens (grass pollen, birch pollen, mixed tree pollens, house dust mite, cat and dog hair, Alternaria and Aspergillus) were also recorded.

### 2.3. Details of the Surgery

All the procedures were performed by the same surgeon (SU). Before treatment, two puffs of co-phenylcaine nasal spray (lidocaine hydrochloride 5% *w*/*v*, phenylephrine 0.5% *w*/*v* and benzalkonium chloride 0.01%) are sprayed into each nostril. Ten minutes later, under endoscopic vision, a rigid nasal endoscopy is performed, and a cotton pledget soaked in adrenaline 1:10,000 is introduced into each nostril. The head of the IT is later injected with Lignospan Special (lidocaine hydrochloride 2% and adrenaline 1:80,000). After 5 min, under endoscopic vision, the radiofrequency wand at a setting of 15 W is introduced into the submucosal IT tissue for approximately 15 s (the exact duration is based on the auto-stop function, which depends on 3D impedance feedback detected by the machine algorithm). This process is repeated in 3 different sites of each IT (anterior, middle and posterior portion). After treatment, a cotton pledget soaked in adrenaline 1:10,000 is left into each nostril, and the patient is asked to wait in the recovery area for post-operative monitoring of vital parameters. After 15 min, the pledgets are removed and the Naseptin cream (chlorhexidine dihydrochloride 0.1% and neomycin sulfate 0.5%) is applied into each nostril. No nasal pack is inserted unless there is an active nosebleed. Patients are discharged without any limitations in their normal daily activities.

### 2.4. Objective and Subjective Measurements at Baseline and Follow-Ups

At T0, T1, T2 and T3, patients underwent objective assessment of nasal airways, smell function and HRQoL, and their subjective sinonasal, olfactory and sleep symptoms were evaluated. All patients also received nasal endoscopy, at both baseline and follow-ups, to evaluate signs of chronic rhinosinusitis (CRS)/rhinitis and post-operative outcomes.

All patients received a home-based sleep test (type III) before being included in the study, and the diagnosis of simple snoring or OSA was established according to the apnoea-hypopnoea index (AHI) calculated from the above-mentioned studies as follows: simple snoring, AHI < 5; mild OSA, 5 ≤ AHI < 15; moderate OSA, 15 ≤ AHI < 30; and severe OSA, AHI ≥ 30. The study was also repeated at 6 months following RFIT.

PNIF and AR were tested on the same occasion to objectively assess patients’ nasal airways. After baseline measurements, a decongestant test was performed using co-phenylcaine (5% lidocaine and 0.5% phenylephrine) topical nasal spray, and measurements were repeated 15 min after its application to reduce any possible influence of the nasal cycle on nasal airflow measurements [27,28,29]. PNIF was measured using a portable Youlten peak flow meter (Clement Clarke International, Mountain Ash, UK). Three maximal inspirations were obtained, and the highest of the three measurements was considered [30]. Unilateral PNIF values were also studied by sealing off one nostril at a time with adhesive tape (Micropore™, 3M™, St Paul, MN, USA), and the highest values were taken as left PNIF (lPNIF) and right PNIF (rPNIF) [31]. AR was tested using an A1 acoustic rhinometer (GM Instruments Ltd., Kilwinning, UK) and conducted while patients held their breath. The minimal cross-sectional area (MCA) and nasal volume (NV) were obtained [32].

The ability to smell was scored using the Sniffin’ Sticks (S’S) 16-item identification test (Burghart, Medisense, Groningen, The Netherlands) [33]. Level of hyposmia was defined as a score below 11 as per normative values reported by Oleszkiewicz and colleagues [33]. Subjective olfactory function was recorded using a visual analogue scale for sense of smell (sVAS—0 indicates “sense of smell absent” and 10 indicates “sense of smell not affected”) [34] and the short version of the Questionnaire of Olfactory Disorders-Negative Statements (short-QODNS) [35].

Other patient-reported outcome measures (PROMs) included the 36-item Short Form Survey (SF-36) used to assess HRQoL, the Sino-Nasal Outcomes Test-22 (SNOT-22) [36] and the Nasal Obstruction Symptom Evaluation (NOSE) to evaluate sinonasal symptoms, as well as the Epworth Sleepiness Scale (ESS) as a subjective measure of patients’ sleepiness.

### 2.5. Statistical Analysis

Quantitative variables were summarized using median and interquartile range (P25–P75), whereas qualitative variables were described with frequency and percentage. Comparisons of measurements between baseline and follow-ups were performed using the paired Wilcoxon test for quantitative variables and the proportion test for dichotomic variables. Ottaviano et al. [37] showed that the relationship between PNIF and covariates is typically not linear, and they proposed a square root transform of PNIF, which has been evaluated as also appropriate for our data. Mixed effect models have been fitted to the data to evaluate the longitudinal effects of the covariates on the studied variables. A goodness-of-fit analysis for each model has been performed using qqplots to validate their use in our study. *p*-values were calculated for all tests, and 5% was considered to be the critical level of significance. All the analyses were performed in R (version 4.4.0, R Core Team, Vienna, Austria, 2021).

## 3. Results

### 3.1. Breakdown of the Population

Seventeen patients were initially included in the study (T0) and underwent RFIT under LA. Thirteen patients attended the 3-month follow-up (T1), fourteen attended the 6-month follow-up (T2) and ten attended the 12-month follow-up (T3).

### 3.2. Demographic Data

The median age of the population was 42.0 years, and there was a higher prevalence of male patients (10; 58.8%). The majority of them were non-smokers (15; 88.2%) and had a history of both snoring and OSA (12; 70.6%). All of the patients complained of bilateral nasal blockages and were unsuccessfully treated medically with nasal douches and steroid sprays (+/− azelastine spray, in cases of allergic rhinitis). Other details of the population, including the patients’ history of previous relevant surgeries of the upper airways as well as comorbidities and routinely taken medications, are reported in Table 1.

### 3.3. Nasal Airflow and Olfactory Function at Baseline

Bilateral and unilateral PNIF values as well as AR parameters pre- and post-decongestion are reported in Table 2. The median identification score at S’S was 13 (Table 2).

### 3.4. Other Investigations at Baseline

The nasal endoscopies confirmed signs of rhinitis and hypertrophy of the IT in all of the cases; in eight patients (47.1%), these were associated with a deviated nasal septum. Skin prick tests confirmed a sensitivity to common aeroallergens in nine patients (52.9%). CT scans of the sinuses showed no concomitant CRS in any of the cases. The majority of the patients were in the moderate OSA category (6; 35.3%) at the time of the pre-operative sleep study with a median oxygen desaturation index (ODI) of 10.5 and a median snore percentage of 24.3%. All of the patients with moderate or severe OSA were using a CPAP machine. The median BMI was 30.1 kg/m^2^ with the majority of the patients (5; 35.7%) being overweight (BMI of 25–29.9) (Table 1, Table 2 and Table 3).

### 3.5. Patient-Reported Outcome Measures (PROMs)

Low median scores on the SF-36 were observed in the domains of energy fatigue (50.0%), general health (60.0%) and health changes (50.0%). The median score for the ESS was 8, that for the SNOT-22 was 31.0, that for the NOSE was 14, that for the short-QODNS was 21 and that for sVAS was 7.5 (Table 3).

### 3.6. Changes at Follow-up

A statistically significant improvement in the patients’ right NV (paired test), left NV and MCA1, as well as their NOSE scores, was demonstrated between T_0_ and T_1_. Apart from that, no other statistically significant changes were observed in the nasal airway parameters either pre- or post-decongestion, S’S scores, BMI, sleep study parameters or PROMs at any of the follow-ups following RFIT (Figure 1, Figure 2 and Figure 3; Table 2 and Table 3).

### 3.7. Influence of Available Variables on Studied Parameters

As seen in the multivariate analysis, left PNIF (pre-decongestion) was significantly negatively influenced by age (older), a higher BMI and the presence of septal deviation. Acoustic rhinometry (pre-decongestion) was significantly negatively influenced by a higher BMI while it was significantly positively influenced by the male sex and the presence of allergic rhinitis. S’S identification was significantly negatively influenced by the presence of septal deviation. The AHI was significantly negatively influenced by the presence of allergic rhinitis, while the NOSE score was significantly negatively influenced by the male sex. The variables influencing the parameters and the strength of these influences are reported in Table 4.

The fitted model, which was created by taking into account the influence of all the available variables, demonstrated a statistically significant improvement of the left NV at 3 and 6 months (*p* = 0.005 and *p* = 0.02, respectively) and at 3 months for the MCA (*p* = 0.005). Similarly, the difference between the baseline and 6-month NOSE scores became statistically significant (*p* = 0.006) in the fitted model (Figure 3). The qqplots analysis confirmed the goodness of fit for each model.

## 4. Discussion

Our prospective study seems to suggest a lack of a significant long-term improvement of nasal airways in patients with SDB following RFIT, with potential benefits, both objective and subjective, limited only to the short-term period (3 and 6 months), as demonstrated by AR and NOSE scores. These findings were further confirmed by our fitted model (Table 2 and Table 3; Figure 2 and Figure 3). The role of RFIT in improving nasal airways is well established in non-SDB patients, although results have often been inconsistent [20]. A systematic review conducted in 2009 on the effectiveness of RFIT confirmed a great variability in the methods used for measuring the subjective relief of nasal blockages [38]. The mean patient-reported nasal obstruction scores decreased statistically significantly in all but one study when the effect of RFIT was measured using VAS scores. Cavaliere et al. [39] demonstrated a significant improvement in the nasal airflow using anterior active rhinomanometry and VAS in a cohort of 25 patients (who had IT hypertrophy refractory to medical treatment), but the decongestion effect significantly decreased at 3 months. On one hand, there is enough evidence to support the use of RFIT in non-SDB patients [23,24,25,26], but on the other hand, its efficacy becomes less obvious when RFIT is evaluated in SDB patients. Casale et al. [40] found a significant reduction in NOSE and VAS scores roughly 45 days following RFIT in patients with simple snoring. The authors also showed an objective significant improvement in nasal airflow using a video-rhino-hygrometer [40]. Means et al. [26], in a retrospective study on 40 patients who underwent RFIT >14 months (14–30 months), which also included eight SDB patients, reported that their relief from nasal obstruction persisted longer than 14 months post-procedure. However, in the only placebo-controlled, double-blind study conducted on SDB patients [41], there was no significant difference in the nasal obstruction outcome as measured by VAS scores, although there was a statistically significant improvement in self-reported CPAP adherence.

The disappointing absence of long-term nasal airways improvement observed in our data is, however, shared by similar studies which evaluated nasal surgery alone in SDB patients [42,43,44]. In fact, both our data and fitted model, the latter taking into account the effect of the available variables on the studied parameters, demonstrated a statistically significant improvement of nasal airways for AR but only in the short-term (either 3 or 6 months) with these changes found to be non-significant at the 12-month follow-up (Table 2 and Table 3; Figure 2 and Figure 3). Similarly, a statistically significant reduction in NOSE scores was demonstrated only in the short-term follow-up at either 3 or 6 months (Figure 3). The NOSE questionnaire is a brief, validated, disease-specific instrument designed to measure nasal obstruction, which has also been confirmed to be a helpful screening tool for OSA [45]. Differently from the SNOT-22, which is more specific for CRS, it does not contain additional questions on otologic, sinus or emotional symptoms. In this regard, the NOSE questionnaire is more specific for nasal obstruction and, thus, able to detect changes in perceived nasal blockage than the SNOT-22. This may suggest that RFIT can actually have a role in improving nasal airways in patients with SDB. Moreover, PNIF may not be the best tool to assess nasal airways in patients with SDB and nasal obstructions, as previously noted [46], and other factors, mainly an altered pharyngeal morphology, [43] could affect the performance of the test and impact the values measured. In this regard, Morinaga and colleagues [43] observed that a favourable nasal surgical outcome in SDB patients was seen in individuals who had a high-positioned soft palate and/or in those with a wide retroglossal space.

The influence of nasal surgery on sleep parameters is not clear, and results are conflicting [47,48,49,50,51]. Although we observed a reduction in the median AHI (−1.3 events/hour), ODI (−1.3 events/hour) and snore percentage (−11.5%) 6 months after RFIT, as well as the halving of the patients’ post-operative ESS scores at 6 and 12 months, none of these were statistically significant (Table 3). Interestingly, a statistically significant negative influence of the presence of non-allergic rhinitis on the AHI was showed in the multivariate analysis (Table 4). According to findings in the literature, surgical success has been defined as a greater than 50% reduction in the AHI and a final AHI of less than 20 [52]. A recent systematic review and meta-analysis on the topic conducted by Schoustra and colleagues [53] revealed a small overall decrease in the AHI of 4.08 events/hour from pre-operative to post-operative sleep study tests. Equally, a previous meta-analysis by Wu and colleagues [54] looking at the effect of isolated nasal surgery on sleep parameters showed a similar mean improvement in AHI of 4.15 events/hour. Overall, these data suggest that nasal surgery alone has a small effect in lowering the AHI, and our data seem to corroborate this. Therefore, taking into account that nasal surgery, including RFIT, appears to not significantly improve sleep parameters, most authors seem to agree on the fact that its benefit in OSA patients could rely on the reduction in the CPAP pressure, which translates into a better CPAP adherence [41,51]. However, even in this aspect, the results are not univocal [26]. In our cohort, all the patients who were using CPAP pre-operatively and kept using it in the follow-up period reported a better adherence to CPAP following RFIT.

Olfactory dysfunction is commonly observed in patients with SDB [55], and older adults with SDB have been reported to be at a higher risk of having impaired odour identification (with an odds ratio of 2.13) [56]. In our study, 23.5% of patients were found to be hyposmic in the identification test. However, although we observed a reduction in the percentage of hyposmic patients during the follow-up period, this was not statistically significant, and this apparent reduction could have been influenced instead by an attrition bias. Similarly, no statistically significant improvement in the reported smell function was observed when looking at their sVAS or short-QODNS scores. OD in SDB patients seems to be related to sleep fragmentation and chronic intermittent hypoxia, causing alterations in the main olfactory bulb neural network and affecting pathways in the central nervous system which involve chemosensory processing [56,57,58]. As a confirmation of this, CPAP therapy has been shown to improve olfactory function [59]. Despite its high prevalence in this population, olfactory function is not frequently assessed when evaluating changes following nasal surgery in SDB patients. Anecdotally, the improvement of olfactory function following RFIT has been documented in non-SDB patients [25,60] but studies looking at patients with SDB are scarce.

HRQoL is impaired in patients with SDB [61], and our results confirmed this with lower scores observed for the SF-36 domains of energy/fatigue, pain, general health and health changes when compared to UK normative values [62]. However, no statistically significant changes were noted in any of the SF-36 domains following RFIT during the follow-up period. In the study of Nilsen and colleagues [63] that included, amongst others, patients with SDB, a significant improvement was demonstrated in the general health and vitality domains of the SF-36 following RFIT. However, they observed that patients with sleep apnoea had poorer outcome after surgery than the other patients [63]. To the best of our knowledge, no studies have yet evaluated the general aspects of HRQoL purely in SDB patients undergoing RFIT; therefore, we were unable to compare our results with those from other authors.

Our multivariate analysis confirmed that several patient-related variables can influence objective and subjective outcomes following RFIT, and these should be taken into account in the patient selection process (Table 4). Final nasal airway measurements, in fact, can be negatively influenced by the presence of a septal deviation, a higher BMI and age (older); interestingly, male patients or those with non-allergic rhinitis may have better results [64]. Similarly, these variables can also affect recorded PROMs. Finally, it is interesting to note that the presence of a septal deviation can also negatively influence olfaction, which is something that has already been confirmed by our research group in previous studies [65,66,67].

### Strengths and Limitations

To the best of our knowledge, this is the only study currently available in the literature in which the effectiveness of RFIT has been evaluated in SDB patients only using multiple objective and subjective outcomes, including olfactory performance, which is often overlooked. Our multivariate analysis and fitted model highlighted multiple variables that can potentially influence recorded outcomes; thus, it can help surgeons improve patient selection when offering RFIT to SDB patients. However, our study is limited by a small sample size; as this can cause erroneous inferences, our results should be carefully interpreted in view of this limitation. Moreover, the addition of a control group (no treatment) to compare our results against could have helped in distinguishing the specific effects of RFIT treatment.

## 5. Conclusions

Patients with SDB frequently experience nasal obstruction, and RFIT can be considered an option for patients with nasal blockage refractory to medical treatment. Our study confirms that the benefits of RFIT alone in SDB patients are limited potentially only to the short-term period. This could be due to the fact that other patient-related variables, including age, sex, BMI and the presence of septal deviation, as well as anatomical factors, like pharyngeal morphology [43], could impact the final outcome. Nevertheless, our results should be confirmed in future studies conducted in larger populations.

## Figures and Tables

**Figure 1 diagnostics-14-01820-f001:**
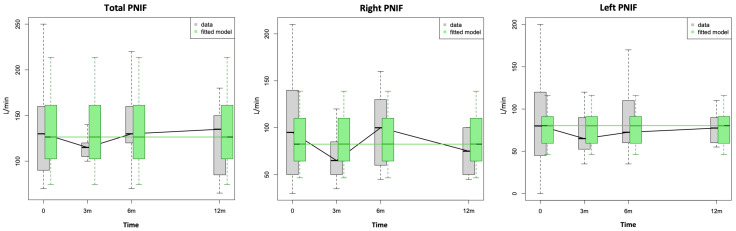
Box-plots showing distribution of total, right and left peak nasal inspiratory flow (PNIF) values at baseline (0) and at 3, 6 and 12 months following radiofrequency of inferior turbinates. The green fitted model was created by taking into account the influence of available variables.

**Figure 2 diagnostics-14-01820-f002:**
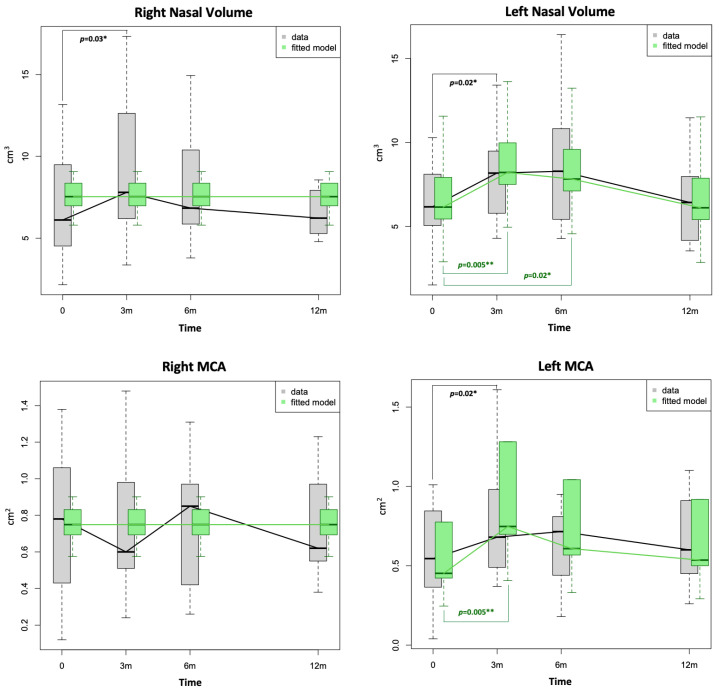
Box-plots showing distribution of right and left nasal volume and right and left minimal cross-sectional area (MCA) values at baseline (0) and at 3, 6 and 12 months following radiofrequency of inferior turbinates. The green fitted model was created by taking into account the influence of available variables. Note that differences in grey refer to the data whilst those in green refer to the fitted model. Level of significance according to *p*-values: * *p* ≤ 0.05, ** *p* ≤ 0.01.

**Figure 3 diagnostics-14-01820-f003:**
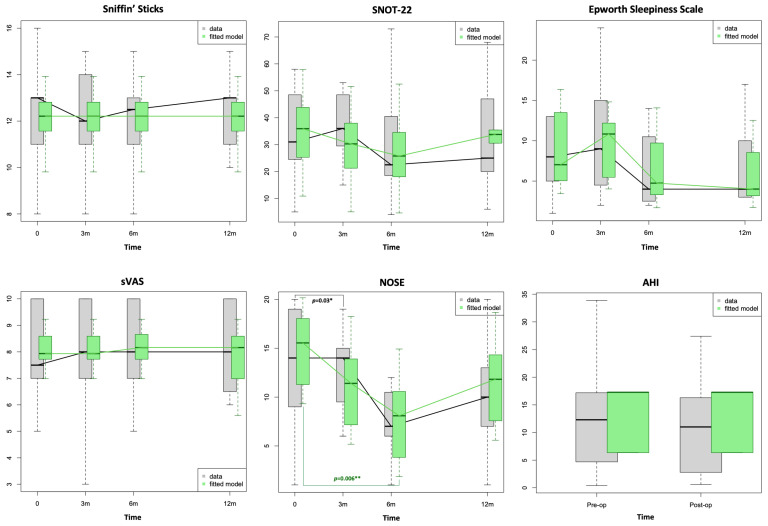
Box-plots showing distribution of patient-reported outcome measures values at baseline (0) and at 3, 6 and 12 months following radiofrequency of inferior turbinates. The green fitted model was created by taking into account the influence of available variables. Note that differences in grey refer to the data whilst those in green refer to the fitted model. SNOT-22: Sino-Nasal Outcomes Test-22; sVAS: Visual Analogue Scale for Smell; NOSE: Nasal Obstruction Symptom Evaluation; AHI: Apnoea–Hypopnea Index. Level of significance according to *p*-values: * *p* ≤ 0.05, ** *p* ≤ 0.01.

**Table 1 diagnostics-14-01820-t001:** General characteristics of the population.

	*n* = 17
** *Demographics* **	
Age, median [P25–P75], year	42.0 [35.0–52.0]
Sex, No (%)	
Female	7 (41.2%)
Male	10 (58.8%)
Smoking status, No (%)	
Ex-smoker	1 (5.9%)
Active	1 (5.9%)
No	15 (88.2%)
History of rhinitis, No (%)	
Allergic type	9 (52.9%)
Non-allergic type	8 (47.1%)
Sleep symptoms, No (%)	
Snoring only	5 (29.4%)
OSA only	0 (0.0%)
Both	12 (70.6%)
Comorbidities, No (%)	
None	7 (41.2%)
Asthma	4 (23.5%)
Hypertension	3 (17.6%)
Mental health issues	3 (17.6%)
Other	5 (29.4%)
Medications, No (%)	
Nasal douche	17 (100%)
Steroid spray	8 (47.1%)
Steroid + antihistamine spray	9 (52.9%)
Sartan	2 (11.8%)
Beta-2 agonist inhaler	4 (23.5%)
Other	5 (29.4%)
Previous relevant surgery, No (%)	
Tonsillectomy	3 (17.6%)
Palatoplasty	2 (11.8%)
Rhinoplasty	2 (11.8%)
Septoplasty	1 (5.9%)
** *Investigations* **	
Skin prick test, No (%)	
Negative	8 (47.1%)
One allergen	3 (17.6%)
Two allergens	2 (11.8%)
Multiple allergens	4 (23.5%)
Nasal endoscopy findings, No (%)	
Rhinitis	17 (100%)
IT hypertrophy only	9 (52.9%)
Septal deviation + IT hypertrophy	8 (47.1%)

OSA: obstructive sleep apnoea; IT: inferior turbinate.

**Table 2 diagnostics-14-01820-t002:** Nasal measurements at baseline and at 3, 6 and 12 months following radiofrequency of inferior turbinates. Significant *p*-values are in bold. Levels of significance * *p* ≤ 0.05.

	Baseline (T_0_) *n* = 17	3-Month (T_1_) *n* = 13	6-Month (T_2_) *n* = 14	12-Month (T_3_) *n* = 10	*p*-Value (T_0_–T_1_)	*p*-Value (T_0_–T_2_)	*p*-Value (T_0_–T_3_)
**Nasal measurements**							
** *Pre-decongestion* **							
PNIF, median [P25–P75], L/min							
Bilateral PNIF	130.0 [90.0–160.0]	115.0 [107.5–120.0]	130.0 [120.0–157.5]	135.0 [93.8–147.5]	0.80	0.83	0.83
Right PNIF	95.0 [50.0–140.0]	65.0 [50.0–82.5]	100.0 [60.0–125.0]	75.0 [52.5–98.8]	0.50	0.70	0.68
Left PNIF	80.0 [45.0–120.0]	65.0 [53.8–85.0]	72.5 [60.0–107.5]	77.5 [61.3–88.8]	0.66	0.92	0.76
Acoustic rhinometry, median [P25–P75]							
Right MCA1, cm^2^	0.8 [0.4–1.1]	0.6 [0.5–1.0]	0.9 [0.5–1.0]	0.6 [0.6–0.9]	0.50	0.85	0.28
Right nasal volume (0–5), cm^3^	6.1 [4.5–9.5]	7.8 [6.2–12.6]	6.8 [5.9–10.0]	6.2 [5.4–7.8]	**0.03 ***	0.90	0.49
Left MCA1, cm^2^	0.5 [0.4–0.8]	0.7 [0.5–1.0]	0.7 [0.5–0.8]	0.6 [0.5–0.9]	**0.02 ***	0.19	0.82
Left nasal volume (0–5), cm^3^	6.2 [5.1–8.1]	8.2 [5.8–9.5]	8.3 [5.7–10.5]	6.4 [4.2–8.0]	**0.02 ***	0.09	0.50
** *Post-decongestion* **							
PNIF, median [P25–P75], L/min							
Bilateral PNIF	150.0 [110.0–180.0]	120.0 [110.0–170.0]	150.0 [125.0–200.0]	140.0 [122.5–155.0]	0.58	0.72	0.72
Right PNIF	110.0 [85.0–130.0]	75.0 [60.0–110.0]	80.0 [70.0–135.0]	97.5 [70.0–128.8]	0.69	0.47	0.26
Left PNIF	100.0 [50.0–140.0]	85.0 [65.0–100.0]	85.0 [60.0–110.0]	100.0 [76.3–100.0]	0.72	0.46	1.00
Acoustic rhinometry, median [P25–P75]							
Right MCA1, cm^2^	1.0 [0.8–1.5]	0.9 [0.8–1.1]	0.9 [0.8–1.3]	1.1 [0.7–1.3]	0.79	0.54	0.37
Right nasal volume (0–5), cm^3^	9.4 [6.0–11.9]	9.3 [7.7–10.9]	8.2 [7.1–11.2]	8.7 [7.3–10.5]	0.24	0.95	0.84
Left MCA1, cm^2^	0.9 [0.6–1.1]	1.0 [0.5–1.2]	1.0 [0.9–1.1]	1.1 [0.9–1.1]	0.19	0.13	0.23
Left nasal volume (0–5), cm^3^	9.5 [6.6–11.6]	9.1 [6.3–12.1]	10.1 [6.2–10.7]	9.9 [8.5–12.0]	0.78	0.79	1.00

PNIF; peak nasal inspiratory flow; MCA1: first minimal cross-sectional area.

**Table 3 diagnostics-14-01820-t003:** Other investigations and patient-reported outcome measures (PROMs) at baseline and at 3-, 6- and 12-month following radiofrequency of inferior turbinates. Significant *p*-values are in bold.

	Baseline (T_0_) *n* = 17	3 Month (T_1_) *n* = 13	6 Month (T_2_) *n* = 14	12 Month (T_3_) *n* = 10	*p*-Value (T_0_–T_1_)	*p*-Value (T_0_–T_2_)	*p*-Value (T_0_–T_3_)
** *Other measurements* **							
Sniffin’ Sticks Identification, median [P25–P75]	13.0 [11.0–13.0]	12.0 [11.0–14.0]	12.5 [11.3–13.0]	13.0 [11.0–13.0]	0.63	0.93	0.10
Normosmics, *n* (%)	13 (76.5%)	11 (84.6%)	12 (85.7%)	9 (90.0%)	0.38	0.20	0.10
Hyposmics, *n* (%)	4 (23.5%)	2 (15.4%)	2 (14.3%)	1 (10.0%)	1.00	0.37	N/A ^+^
BMI, median [P25–P75], kg/m^2^	30.1 [26.5–32.8]	-	27.1 [25.5–32.0]	-	-		-
Normal range (18.5–24.9), *n* (%)	2 (14.3%)		3 (23.1%)			0.47	
Overweight, (25–29.9), *n* (%)	5 (35.7%)		5 (38.5%)			0.93	
Obese grade I, (30–34.9), *n* (%)	4 (28.6%)		3 (23.1%)			1	
Obese grade II, (35–39.9), *n* (%)	2 (14.3%)		1 (7.7%)			1	
Obese grade III, (≥40), *n* (%)	1 (7.1%)		1 (7.7%)			1	
Missing	3		4			1	
Sleep Study		-		-	-		-
AHI, median [P25–P75]	12.3 [4.7–17.2]		11.0 [2.8–16.3]			0.42	
Normal (<5), *n* (%)	5 (29.4%)		5 (33.3%)			1	
Mild OSA (5–14.9), *n* (%)	5 (29.4%)		5 (33.3%)			1	
Moderate OSA, (15–29.9), *n* (%)	6 (35.3%)		4 (26.7%)			0.89	
Severe OSA (≥30), *n* (%)	1 (5.9%)		1 (6.7%)			1	
ODI, median [P25–P75]	10.5 [3.7–14.6]		9.2 [2.4–14.3]			0.48	
Snore percentage, median [P25–P75]	24.3 [5.6–36.5]		13.8 [2.0–29.7]			0.89	
Missing	0		2				
** *PROMs* **							
SF-36, median [P25–P75], %							
Physical functioning	90.0 [80.0–100]	85.0 [60.0–95.0]	90.0 [70.0–100]	90.0 [85.0–100]	0.15	0.94	0.41
Role limitations due to physical health	100 [25.0–100]	100 [62.5–100]	100 [75.0–100]	100 [50.0–100]	0.58	0.34	1.00
Role limitations due to emotional problems	100 [33.3–100]	100 [50.0–100]	100 [100–100]	100 [100–100]	0.79	0.09	0.37
Energy/Fatigue	50.0 [45.0–65.0]	45.0 [42.5–65.0]	45.0 [35.0–70.0]	50.0 [40.0–65.0]	0.73	0.97	0.83
Emotional wellbeing	80.0 [56.0–88.0]	76.0 [62.0–76.0]	76.0 [64.0–84.0]	84.0 [48.0–88.0]	0.93	0.30	0.32
Social functioning	81.3 [50.0–90.6]	75.0 [50.0–75]	75.0 [62.5–100]	87.5 [62.5–100]	0.26	0.55	0.46
Pain	78.8 [45.0–82.5]	77.5 [61.3–95.0]	77.5 [57.5–90.0]	67.5 [67.5–77.5]	0.41	0.76	0.17
General health	60.0 [40.0–70.0]	60.0 [35.0–65.0]	65.0 [55.0–75.0]	65.0 [35.0–70.0]	0.33	0.08	0.80
Health change	50.0 [25.0–75.0]	50.0 [50.0–62.5]	50.0 [50.0–75.0]	50.0 [50.0–75.0]	0.17	0.85	0.42
Epworth sleepiness scale, median [P25–P75]	8.0 [5.0–13.0]	9.0 [4.5–15.0]	4.0 [2.5–10.5]	4.0 [3.0–10.0]	0.55	0.07	0.20
Short-QODNS, median [P25–P75]	21.0 [15.5–21.0]	20.0 [15.0–21.0]	20.0 [15.3–21.0]	18.5 [15.3–21.0]	0.78	0.85	1.00
sVAS, median [P25–P75]	7.5 [7.0–10]	8.0 [7.0–10]	8.0 [7.0–10]	8.0 [6.5–10.0]	0.34	0.68	0.72
SNOT-22, median [P25–P75]	31.0 [24.5–48.5]	36.0 [29.5–48.5]	22.5 [18.8–40.3]	25.0 [20.0–47.0]	0.49	0.42	0.55
NOSE, median [P25–P75]	14.0 [9.3–18.3]	14.0 [9.5–15.0]	7.0 [6.5–9.8]	10.0 [7.0–13.0]	**0.03 ***	0.09	0.83

Level of significance according to *p*-values: * *p* ≤ 0.05. ^+^ *p*-value unobtainable considering only 1 hyposmic patient is present at T3. PROMs: patient-reported outcome measures; BMI: body mass index; OSA: obstructive sleep apnoea; AHI: Apnoea–Hypopnea Index; ODI: oxygen desaturation index; SF-36: 36-item Short Form Survey; sVAS: Visual Analogue Scale for Sense of Smell; SNOT-22: 22-item SinoNasal Outcome Test; NOSE: Nasal Obstruction and Septoplasty Effectiveness Scale.

**Table 4 diagnostics-14-01820-t004:** Effect of the variables on pre-decongestion nasal airway measurements, patient-reported outcome measures (PROMs) and olfactory test in the multivariate analysis. Level of significance according to *p*-values: * *p* ≤ 0.05, ** *p* ≤ 0.01, *** *p* ≤ 0.001.

	Age	Sex (Male)	BMI	Septal Deviation	Non-Allergic Rhinitis	Random Effect (Patient)
**PNIF**						
Bilateral						1.60
Right						1.34
Left	−0.10 *		−0.22 ***	−2.42 **		
**Acoustic Rhinometry**						
Right MCA1			−0.03 **			
Left MCA1		+0.60 ***			+0.54 ***	
Right NV			−0.03 **			
Left NV		+2.43 *			+2.22 *	1.56
**AHI**					−19.9 ***	
**Epworth sleepiness scale**		−1.30		−2.63	−2.41	4.87
**SNOT-22**		−5.42	+0.92	+1.29	−11.28	13.65
**NOSE**		−6.94 **		+1.01	−4.49	1.30
**sVAS**						1.39
**Sniffin’ Sticks (Identification)**				−2.25 ***		

PNIF; peak nasal inspiratory flow; MCA1: first minimal cross-sectional area; NV: nasal volume; AHI: Apnoea–Hypopnea Index; SNOT-22: 22-item Sinonasal Outcome Test; NOSE: Nasal Obstruction and Septoplasty Effectiveness Scale; sVAS: Visual Analogue Scale for Sense of Smell; BMI: body mass index.

## Data Availability

Data are available upon request to the senior author.

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
