# Peer review of "Objective and Subjective Outcomes Following Radiofrequency of Inferior Turbinates in Patients with Sleep-Disordered Breathing"

_diagnostics, 2024, doi:10.3390/diagnostics14161820_

Round 1
Reviewer 1 Report
Comments and Suggestions for Authors
El estudio aborda un área relativamente poco explorada al centrado tanto en los resultados objetivos como subjetivos del tratamiento con radiofrecuencia de los cornetes inferiores en pacientes con trastornos respiratorios del sueño (TRS). Incluye una variedad de medidas de resultados, como el flujo de aire nasal, la función olfativa y la calidad del sueño.
El concepto de utilizar radiofrecuencia en los cornetes inferiores no es nuevo y se han realizado estudios similares, aunque principalmente en pacientes sin SDB.
El artículo no proporciona información innovadora que pueda alterar significativamente las prácticas o pautas actuales, pero agrega datos útiles para un subconjunto específico de pacientes.
Algunas partes del análisis podrían ser más concisas. Hay algunas repeticiones en la descripción de los resultados del estudio y en las comparaciones con la literatura existente.
La metodología es exhaustiva, con definiciones claras de los parámetros del estudio y de los criterios de inclusión. Llama la atención que no se utilice la técnica de referencia para la evaluación de la función ventilatoria nasal, la rinomanometría anterior activa.
Tamaño de muestra pequeño que puede limitar la generalización de los hallazgos.
El diseño del estudio podría incluir un grupo de control para un análisis comparativo más sólido.
Recomendaría la rinomanometría: el uso de la rinomanometría anterior como técnica para evaluar la resistencia y la ventilación de cada fosa nasal proporcionaría datos objetivos sobre el flujo y la resistencia de cada fosa nasal durante la respiración nasal; Ampliar el tamaño de la muestra : aumentar el número de participantes podría mejorar la solidez y la aplicabilidad de los hallazgos; Grupo de control : incluir un grupo de control sometido a un tratamiento diferente oa ningún tratamiento podría ayudar a distinguir los efectos específicos del tratamiento de radiofrecuencia; Seguimiento más prolongado : ampliar el período de seguimiento podría proporcionar información sobre la eficacia y seguridad a largo plazo del procedimiento.
Author Response
In response to Reviewer #1.
Please note that the reviewer’s comments were in Spanish. We translated these using Google translate, but it may contain errors. We hope the translation was accurate enough.
- “El estudio aborda un área relativamente poco explorada al centrado tanto en los resultados objetivos como subjetivos del tratamiento con radiofrecuencia de los cornetes inferiores en pacientes con trastornos respiratorios del sueño (TRS). Incluye una variedad de medidas de resultados, como el flujo de aire nasal, la función olfativa y la calidad del sueño. El concepto de utilizar radiofrecuencia en los cornetes inferiores no es nuevo y se han realizado estudios similares, aunque principalmente en pacientes sin SDB. El artículo no proporciona información innovadora que pueda alterar significativamente las prácticas o pautas actuales, pero agrega datos útiles para un subconjunto específico de pacientes.” The study addresses a relatively underexplored area by focusing on both objective and subjective outcomes of radiofrequency treatment of the inferior turbinates in patients with sleep-disordered breathing (SDB). It includes a variety of outcome measures, such as nasal airflow, olfactory function, and sleep quality. The concept of using radiofrequency on the inferior turbinates is not new and similar studies have been performed, although mainly in patients without SDB. The article does not provide ground-breaking information that could significantly alter current practices or guidelines, but adds useful data for a specific subset of patients.
- We thank the reviewer for his/her appreciation of our work.
- “Algunas partes del análisis podrían ser más concisas. Hay algunas repeticiones en la descripción de los resultados del estudio y en las comparaciones con la literatura existente. La metodología es exhaustiva, con definiciones claras de los parámetros del estudio y de los criterios de inclusión. Llama la atención que no se utilice la técnica de referencia para la evaluación de la función ventilatoria nasal, la rinomanometría anterior activa.Tamaño de muestra pequeño que puede limitar la generalización de los hallazgos. El diseño del estudio podría incluir un grupo de control para un análisis comparativo más sólido. Recomendaría la rinomanometría: el uso de la rinomanometría anterior como técnica para evaluar la resistencia y la ventilación de cada fosa nasal proporcionaría datos objetivos sobre el flujo y la resistencia de cada fosa nasal durante la respiración nasal; Ampliar el tamaño de la muestra : aumentar el número de participantes podría mejorar la solidez y la aplicabilidad de los hallazgos; Grupo de control : incluir un grupo de control sometido a un tratamiento diferente oa ningún tratamiento podría ayudar a distinguir los efectos específicos del tratamiento de radiofrecuencia; Seguimiento más prolongado : ampliar el período de seguimiento podría proporcionar información sobre la eficacia y seguridad a largo plazo del procedimiento.” Some parts of the analysis could be more concise. There is some repetition in the description of the study results and in comparisons with existing literature. The methodology is exhaustive, with clear definitions of the study parameters and inclusion criteria. It is striking that the reference technique for the evaluation of nasal ventilatory function, active anterior rhinomanometry, is not used. The small sample size may limit the generalizability of the findings. The study design could include a control group for a more robust comparative analysis. I would recommend rhinomanometry:
- Using anterior rhinomanometry as a technique to assess the resistance and ventilation of each nostril would provide objective data on the flow and resistance of each nostril during nasal breathing;
- Expand sample size: Increasing the number of participants could improve the robustness and applicability of the findings;
- Control group: Including a control group undergoing a different treatment or no treatment could help distinguish the specific effects of the radiofrequency treatment;
- Longer follow-up: Extending the follow-up period could provide information on the long-term efficacy and safety of the procedure
- We thank the reviewer for his/her comments.
- We acknowledge that the lack of anterior active rhinomanometry as the tool to assess unilateral nasal resistance and airflow could potentially be seen a limitation of the study. However, from previous research on comparison of different tools to assess nasal airflow, we found a significant moderate correlation between PNIF and anterior active rhinomanometry (AAR) with only PNIF showing a significant correlation (weak-moderate) with the reported nasal symptom scores. (Ottaviano G, Pendolino AL et al, 2022) This good correlation between PNIF and AAR was also observed when studying the nasal cycle, with PNIF offering a lower variability compared to AAR (Pendolino et al, 2018; Pendolino et al, 2019). These results seem to suggest that these two techniques are pretty much interchangeable when used to measure nasal airflow, although PNIF seems to be more reliable. Nevertheless, after a careful PubMed search, we found only one paper on SDB and RFIT using AAR (Oyake et al, 2004) which, however, is in Japanese and not included in our paper. All the studies mentioned in the manuscript used different tools as outcome measures. Therefore, this does not constitute a true study limitation of the study as it does not limit the comparisons with other available studies on the same topic.
- We agree with the author that a longer follow-up could have provided more information on the duration of benefits following RFIT. However, we should remember this was a real-life study and, in a busy national health system, patients are generally discharged to their general practitioner after an average time of 3-6 months, if no additional issues are observed and/or if long-term medical treatment is required. Nevertheless, a follow-up of 12 months is widely accepted as being a long-term follow-up in research. In the systematic review on RFIT for patients with nasal symptoms (Hytönen et al, 2009 – number 38 in the manuscript), out of 35 papers included, only 5 (14.3%) had a follow-up longer than 12 months; in fact, the majority had a follow-up ≤ 10 months (19/35; 54.3%). Therefore, we believe a 12-month follow-up could be considered a reasonable length of follow-up for our study.
- The small sample size is a limitation of the study and we already acknowledged this in the section “Strengths and limitations” at the end of the discussion. A lack of attendance of the patients at their follow-ups during the study period, has definitely impacted on our data analysis, as mentioned. We agree with the author that a larger sample size could have improved the robustness and applicability of the findings observed.
- Finally, the addition of a control group (probably with no treatment) could have helped in distinguishing the specific effects of RFIT treatment. Unfortunately, we do not have a historic control group with similar characteristics we could include to potentially compare our results with. We added this amongst the limitations of the study. Thanks for pointing this out, as it could be the object of a future study moving forward. (Lines 425-426)
References:
- Ottaviano G, Pendolino AL, Scarpa B, Torsello M, Sartori D, Savietto E, Cantone E, Nicolai P. Correlations between Peak Nasal Inspiratory Flow, Acoustic Rhinometry, 4-Phase Rhinomanometry and Reported Nasal Symptoms. J Pers Med. 2022 Sep 15;12(9):1513.
- Pendolino AL, Nardello E, Lund VJ, Maculan P, Scarpa B, Martini A, Ottaviano G. Comparison between unilateral PNIF and rhinomanometry in the evaluation of nasal cycle. Rhinology. 2018 Jun 1;56(2):122-126.
- Pendolino AL, Scarpa B, Ottaviano G. Relationship Between Nasal Cycle, Nasal Symptoms and Nasal Cytology. Am J Rhinol Allergy. 2019 Nov;33(6):644-649.
- Oyake D, Ochi K, Takatsu M, Shintani T, Umehara T, Koizuka I. [Clinical effect of bipolar radiofrequency thermotherapy on allergic rhinitis]. Nihon Jibiinkoka Gakkai Kaiho. 2004 Jul;107(7):695-701. Japanese
- Hytönen ML, Bäck LJ, Malmivaara AV, Roine RP. Radiofrequency thermal ablation for patients with nasal symptoms: a systematic review of effectiveness and complications. Eur Arch Otorhinolaryngol. 2009 Aug;266(8):1257-66.
Reviewer 2 Report
Comments and Suggestions for Authors
In this real-life prospective cohort study Pendolino et al examine Objective and subjective outcomes following radiofrequency of inferior turbinates in patients with sleep-disordered breathing.
My major comment is that lack of any associations may be simply due to low number of patients. There are multiple comparisons of various outcomes, without clearly defined primary and secondary outcomes which can cause multiple comparisons fallacy. These limitations should be clearly described in a separate paragraph at the end of discussion section. Nevertheless, this exploratory analysis may be of interest despite these limitations as it evaluates RFIT in SDB prospectively over 12 months with objective evaluation.
My other comments are as follows:
1. Please use OSA insted of "apnea" and "apnea only" troughout the manuscript. Some patients may have only hypopnea on polysomnography and still have OSA. How was OSA diagnosed polygraphy, or full PSG? Did the outcomes differ in patients with snoring only and those with both?
2. Statistics: paired analysis should be used as the authors measure improvements in same patient after intervention.
3. Revising discussion so that the first paragraph that summarizes main findings of your research. This may improve clarity of the manuscript.
4. Results paragraph 3.7. Again, limited number of patients precludes meaningful statistical analysis and may lead to erroneous inferences. Building a model on 17 patients does not make much sense. I would suggest deleting this section and revising the manuscript accordingly.
5. Conclusion: "RFIT may represent an important adjunct in improving CPAP tolerance in OSA patients."-you have not evaluated this so this is a speculation. Please delete.
6. Finally, the authors may want to consult a statistician based on comments above
Author Response
In response to Reviewer #2:
- In this real-life prospective cohort study Pendolino et al examine Objective and subjective outcomes following radiofrequency of inferior turbinates in patients with sleep-disordered breathing. My major comment is that lack of any associations may be simply due to low number of patients. There are multiple comparisons of various outcomes, without clearly defined primary and secondary outcomes which can cause multiple comparisons fallacy. These limitations should be clearly described in a separate paragraph at the end of discussion section. Nevertheless, this exploratory analysis may be of interest despite these limitations as it evaluates RFIT in SDB prospectively over 12 months with objective evaluation.
- We thank the reviewer for his/her comment. The small sample size definitely constitutes a limitation of the study and this has been already mentioned few times in the text including in the specific section “strength and limitations”, which was already present in the original version of the manuscript. (Lines 351-352 and 423-424).
- We agree on the lack of a clear definition of the primary and secondary outcomes. These have now been defined in the text. (Lines 88-93)
- Please use OSA instead of "apnea" and "apnea only" throughout the manuscript. Some patients may have only hypopnea on polysomnography and still have OSA. How was OSA diagnosed polygraphy, or full PSG? Did the outcomes differ in patients with snoring only and those with both?
- We thank the reviewer for his/her comments. We agree with the reviewer on the use of the term OSA instead of "apnea" and "apnea only". This has now been changed throughout the text and in the Tables.
- Apologies on the lack of information about the type of sleep study performed. This information has now been added in the text. (Lines 130-134)
- Outcomes did not differ in patients with snoring only and those with apnoea only or both. The only variables shown to have an influence at the multivariate analysis on the studied parameters are those reported in the Section 3.7. (Lines 281-308) Table 4 has also been added to make clearer the effects of the variables on the studied parameters. (Table 4)
- Statistics: paired analysis should be used as the authors measure improvements in same patient after intervention.
- We thank the reviewer for his/her comment. We apologise for the inattention. This has now been corrected (Line 160) and values in Tables 2-3 have been corrected as a consequence. (Tables 2-3) However, the outcomes did not change and, therefore, this mistake did not impact on the whole paper/discussion/conclusions.
- Revising discussion so that the first paragraph that summarizes main findings of your research. This may improve clarity of the manuscript.
- We thank the reviewer for his/her comment. We understand the reviewer’s point and we decided to rearrange the discussion in a way that the main findings are now at the beginning of each paragraph of the discussion. As suggested, this may improve clarity of the manuscript. However, we decided to not create a separate paragraph to summarise all the results, to avoid the discussion to seem fragmented and disconnected with the rest of the text when comparing our findings against the literature. (Lines 310-426)
- Results paragraph 3.7. Again, limited number of patients precludes meaningful statistical analysis and may lead to erroneous inferences. Building a model on 17 patients does not make much sense. I would suggest deleting this section and revising the manuscript accordingly.
- We thank the reviewer for his/her comment. We agree with the reviewer that a limited number of patients can lead to a large variability on the estimates. However, a model can help in cases where little information is present with regard to the data available. For this reason, we preferred to keep the section on the model in our paper. In order to validate their use in our study, we performed a goodness of fit analysis for each model. If the model fits well the data distribution, then the plotted points are disposed on a diagonal. In our case all the models essentially fit well our data, despite a small sample size. (See figure attached “QQPlots”) Therefore, the qqplots showed that the distribution hypothesis is satisfied for each model. This picture, however, has not been added in the manuscript to avoid further confusion for the readers and we decided, instead, to add few sentences in the statistical analysis and results to summarise that. (Lines 165-166, 307-308)
- Conclusion: "RFIT may represent an important adjunct in improving CPAP tolerance in OSA patients."-you have not evaluated this so this is a speculation. Please delete.
- We thank the reviewer for his/her comment. This has now been deleted. (Lines 433-435)
- Finally, the authors may want to consult a statistician based on comments above
- We thank the reviewer for his/her comment. A statistician had already been involved in the data analysis in the first instance. Bruno Scarpa is one of the authors of the paper.

Reviewer 3 Report
Comments and Suggestions for Authors
The followings are my observations and suggestions while reviewing the paper.
1) This research fall under the scope of this journal.
2) The research has significant interest and looks very promising in most of the part.
3) The method needs more elaboration in terms of novelty with state of the art methods.
4) Results need more metric to validate. Along with box plot please include other plotting methods like ROC curve or others.
Author Response
In response to Reviewer #3:
The followings are my observations and suggestions while reviewing the paper.
1) This research falls under the scope of this journal.
2) The research has significant interest and looks very promising in most of the part.
3) The method needs more elaboration in terms of novelty with state-of-the-art methods.
4) Results need more metric to validate. Along with box plot please include other plotting methods like ROC curve or others.
- We thank the reviewer for his/her appreciation on our work.
- We are unsure on what the reviewer meant with “more elaboration in terms of novelty with state-of-the-art methods” and do not understand whether this is a simple statement or there is something that needs to be done from our side. We believe this section has been thoroughly described but we are happy to revise that if the reviewer provides a more analytic comment. Nevertheless, a section on sleep study and OSA classification has been added in the methods section, as requested by the other reviewer. Hopefully, this could also answer this reviewer’s request. (Lines 130-134)
- All the variables considered are quantitative, so the ROC curve, which is very useful for classification tasks, cannot be obtained. In order to validate the use of models in our study, we performed a goodness of fit analysis for each model. If the model fits well the data distribution, then the plotted points are disposed on a diagonal. In our case all the models essentially fit well our data, despite a small sample size. (See figure QQPlots attached) Therefore, the qqplots showed that the distribution hypothesis is satisfied for each model. This picture, however, has not been added in the manuscript to avoid further confusion for the readers and we decided, instead, to add few sentences in the statistical analysis and results to summarise that. (Lines 165-166, 307-308) Additionally, Table 4 has also been added to make clearer the effects of the variables on the studied parameters. (Table 4)

Round 2
Reviewer 1 Report
Comments and Suggestions for Authors
The authors have responded satisfactorily to the suggestions and queries raised in the previous review.
Thank you
Author Response
- The authors have responded satisfactorily to the suggestions and queries raised in the previous review. Thank you.
- We thank the reviewer for his/her comment.
Reviewer 2 Report
Comments and Suggestions for Authors
The manuscript is somewhat improved, however there are multiple comparisons of various outcomes outside primary aim in only 17 patients that can cause erroneous inferences that a lot of discussion is built on.
Minor: for skewed data non parametric alternative to paired T test should be used (such as Wilcoxon signed-rank test) although this will also give non-significant result as well.
Comments on the Quality of English Language
.
Author Response
- The manuscript is somewhat improved. However, there are multiple comparisons of various outcomes outside primary aim in only 17 patients that can cause erroneous inferences that a lot of discussion is built on. Minor: for skewed data non parametric alternative to paired T test should be used (such as Wilcoxon signed-rank test) although this will also give non-significant result as well.
- We thank the reviewer for his/her comment. The analysis has now been repeated using the Wilcoxon test as the reviewer suggested. This led to little changes in the results and these have been now commented in the text. (Edits highlighted in yellow) However, limitations of running multiple tests on the same data set at the same stage of an analysis should be also taken into account. Nevertheless, the main conclusions of the study have not changed. Our study is limited by a small sample size and as the reviewer mentioned this can cause erroneous inferences. Although this was already mentioned in the “strengths and limitations” section, this limitation has been further highlighted in the text.
Round 3
Reviewer 2 Report
Comments and Suggestions for Authors
In revised manuscript and table 3 the authors report improvement in NOSE at T1 that was not reported before. But when you look at the number seems there is no difference between T0 and T1 14.0 [9.3-18.3] vs. 14.0 [9.5-15.0] and is actually the smallest difference comparing to other visits where there was no statistical significance.
I suggest the authors consult a statistician and carefully proofread the manuscript explaining any changes in findings from previous version.
Author Response
- In revised manuscript and table 3 the authors report improvement in NOSE at T1 that was not reported before. But when you look at the number seems there is no difference between T0 and T1 14.0 [9.3-18.3] vs. 14.0 [9.5-15.0] and is actually the smallest difference comparing to other visits where there was no statistical significance. I suggest the authors consult a statistician and carefully proofread the manuscript explaining any changes in findings from previous version.
We thank the reviewer for his/her comment. A statistician had already been involved in the data analysis from the beginning and assisted throughout the revision process. Bruno Scarpa is one of the authors of the paper and has further reviewed the results upon the reviewer’ request.
The paired Wilcoxon test (as previously suggested from the reviewer), has been run again for the difference between T0-T1 for the NOSE score, as requested. We can confirm the test is significant, p=0.03. Medians are equal at T0 and T1 as reported in Table 3. However, the reason why the p is significant between T0 and T1 for NOSE, despite similar median values, is because we ran a ‘paired’ test, which compares the differences in NOSE score for each single patient between T0 and T1. In fact, the median of the differences in the scores between T0and T1 within each individual is -2 (for the majority of the patients the NOSE score diminishes between T0 and T1). This is different from comparing the median values of NOSE score at T0 and at T1 which, instead, represent only the median of the NOSE scores for the whole group obtained either at T0 or at T1. This has now been highlighted in the text (Line 249)